# A modular protein language modelling approach to immunogenicity prediction

Hugh O'Brien[1], Max Salm[1]*, Laura T. Morton[1], Maciej Szukszto[2], Felix O'Farrell[1], Charlotte Boulton[2], Laurence King[1,2], Supreet Kaur Bola[2], Pablo D. Becker[1], Andrew Craig[1], Morten Nielsen[3], Yardena Samuels[4], Charles Swanton[5], Marc R. Mansour[2,6], Sine Reker Hadrup[3]*, Sergio A. Quezada[1,2‡]*

**1** Achilles Therapeutics UK Ltd, United Kingdom, **2** Research Department of Haematology, UCL Cancer Institute, University College London, London, United Kingdom, **3** Technical University of Denmark, Lyngby, Denmark, **4** Weizmann Institute of Science, Rehovot, Israel, **5** The Francis Crick Institute, London, United Kingdom, **6** Department of Developmental Biology and Cancer, Great Ormond Street Institute of Child Health, UCL

‡ on behalf of the DECOD-Ag Consortium.
* m.salm@achillestx.com (MS); sirha@dtu.dk (SRH); s.quezada@ucl.ac.uk (SAQ)

**Data Availability Statement:** All data used in this study are available from public data sources and other publications. VDJdb is available at https://vdjdb.cdr3.net/; CEDAR is available at https://cedar.iedb.org/; McPas-TCR is available at https://

## Abstract

Neoantigen immunogenicity prediction is a highly challenging problem in the development of personalised medicines. Low reactivity rates in called neoantigens result in a difficult prediction scenario with limited training datasets. Here we describe ImmugenX, a modular protein language modelling approach to immunogenicity prediction for CD8+ reactive epitopes. ImmugenX comprises of a pMHC encoding module trained on three pMHC prediction tasks, an optional TCR encoding module and a set of context specific immunogenicity prediction head modules. Compared with state-of-the-art models for each task, ImmugenX's encoding module performs comparably or better on pMHC binding affinity, eluted ligand prediction and stability tasks. ImmugenX outperforms all compared models on pMHC immunogenicity prediction (Area under the receiver operating characteristic curve = 0.619, average precision: 0.514), with a 7% increase in average precision compared to the next best model. ImmugenX shows further improved performance on immunogenicity prediction with the integration of TCR context information. ImmugenX performance is further analysed for interpretability, which locates areas of weakness found across existing immunogenicity models and highlight possible biases in public datasets.

## Author summary

Accurate prediction of neoantigen immunogenicity has the potential to greatly improve the effectiveness of targeted therapies for cancer. While there are a number of associated tasks, such as peptide-HLA elution prediction, which can be now predicted with high accuracy by a number of published models, direct prediction of which epitopes will produce an immune response has proven more difficult. In this paper we demonstrate a modular protein language model approach which is trained iteratively to include data from related sub-tasks and can be extended to include information such as candidate TCRs of

friedmanlab.weizmann.ac.il/McPAS-TCR/; the BigMHC MANAFEST dataset is available at https://github.com/KarchinLab/bigmhc; the NetTCR 2.1 experiment data is available at https://github.com/mnielLab/NetTCR-2.1; the STAPLER experiment dataset is available at https://github.com/NKI-AI/STAPLER. The netMHCstabpan data is available at https://services.healthtech.dtu.dk/suppl/immunology/NetMHCstabpan-1.0/. Code availability: Code to run ImmugenX can be found at https://doi.org/10.5281/zenodo.13850954.

**Funding:** This project has received funding from the European Union's Horizon 2020 research and innovation programme under grant agreement No 964998 awarded to S.A.Q., C.S., M.R.M., Y.S. and S.R.H. The funders had no role in study design, data collection and analysis, decision to publish, or preparation of the manuscript.

interest when available. There is a relatively small amount of immunogenicity data available, with even less data available with paired TCRs. This makes directly training a model to predict immunogenicity challenging. Our approach has the advantage of utilising data from sub-tasks and masked language modelling to allow for training a highly performant model with a small dataset. Using a cancer-specific benchmarking dataset we show this approach improves on existing state-of-the-art models and can be improved further with the addition of TCR context. This provides a framework that can serve as the basis for utilising additional information sources and datasets as they become available.

## 1. Introduction

High-quality in silico prediction of antigen immunogenicity could be a key factor in the development of effective personalised therapies for cancer. Therapies based on tumour derived neoantigens, often specific to that patient, have gained traction in recent years. Correctly selecting target antigens is a critical step in treatment development, since the mutations in a patient's tumour can greatly exceed treatment payloads. Rates of reactivity in peptides can be as low as 2–6% of called mutations [1, 2] for a given patient's tumour when tested. In high mutation settings, this makes the ranking of candidate mutations key to designing an effective therapy. ImmugenX and the models benchmarked in this work focus on HLA class I presented epitopes, recognised by CD8+ T-cells, as the definition of immunogenicity; however, a similar architecture and training approach could in future be applied to class II and CD4+ reactivities.

There are many models focused on predicting prerequisite steps to T-cell recognition. Binding affinity (BA) models such as netMHCpan (3.0 and earlier) [3] and MHCFlurry [4] are trained on quantitative affinity measurements of peptide-MHC (pMHC) complexes. Newer versions of netMHCpan[5], MHCFlurry[6] and comparable models such as MixMHCpred [7] are trained to predict whether peptides are naturally eluted ligands (EL) as measured by mass spectrometry. Several more recent methods have been developed using more complex neural network architectures such as RNNs [8] and transformers [9] to solve these tasks. Efforts have also been made to automatically benchmark existing models as new data becomes available [10], with current findings showing strong agreement between several models with overall high performance. A smaller number of models have been reported that predict other pMHC formation-related metrics such as pMHC stability [11], as measured by dissociation assays, and TAP transport efficiency [12]. Predictions based on these pMHC tasks are all known to correlate with T-cell immunogenicity.

Direct predictors of immunogenicity have also been developed, with many using the above pMHC models as a basis. PRIME uses the MixMHCpred affinity rank, along with likely TCR interacting amino acids from the epitope as input to a neural network [7]. Gartner et al. [13] and Müller et al [14] built machine learning models taking existing pMHC models and engineered features as inputs, including some patient-specific antigenicity features such as expression. Integrated deep learning approaches have also been developed, such as BigMHC, which transfer-learns immunogenicity from its previously trained RNN-based EL model [8]. Due to low volumes of training data and small independent test datasets, these models have generally shown improved performance at the time of publication but some reduced performance on new data as it becomes available. This indicates current models do not yet represent highly generalisable underlying features of immunogenic neoantigens.

TCR specificity is also a relevant task to immunogenicity prediction, especially when considering the application of personalised therapy design. The likelihood of immunogenicity in

the context of a patient's TCR repertoire would be a desirable metric for ranking candidate mutations. TCR specificity models can perform poorly on entirely unseen epitopes after accounting for biases in test dataset composition [15, 16, 17, 18]. Poor generalisation is likely due to a shortage of diverse datasets since the current repositories of paired pMHC-TCR triplets are heavily biased to HLA-A02:01 and a few viral epitopes [19]. While these datasets are currently limited in scope, next-generation screening methods may greatly increase volumes of paired data in the coming years [20], meaning an increase in TCR data from which to retrain existing models. Integrating TCR prediction into an immunogenicity model would allow it to take full advantage of these new datasets.

Here we present ImmugenX a modular immunogenicity prediction protein language model based on the transformer architecture. ImmugenX comprises a pMHC submodule, trained sequentially on multiple pMHC prediction tasks. In the first instance, this provides the input embeddings for an immunogenicity prediction head model to perform pMHC-only immunogenicity prediction. ImmugenX is also extendable to additional patient-specific inputs, by appending embeddings from purpose-trained encoding networks, demonstrated here with TCR sequences. We show that ImmugenX achieves a state-of-the-art level of performance at pMHC immunogenicity on an independent cancer-specific holdout and is extendable to include additional features when they are available.

## 2. Results

### 2.1 ImmugenX is built on a high performance, multi-task Peptide-MHC encoding module

A sub-module of ImmugenX was trained to encode peptide-MHC input pairs for downstream immunogenicity prediction by training iteratively on related pMHC prediction tasks. This allows us to use all available data on immunogenicity related pMHC prediction tasks in the training of ImmugenX. Performance on these sub-tasks was assessed by comparison to state-of-the-art models at each stage. First, the BA task was assessed against MHCFlurry 2.0 [6] on an IEDB [21] search for BA data deposited after 2021, filtered for 8mer similarity to the training set, ensuring a clean test set for both models. Second, using the NetMHCpan 4.1 single-allelic eluted ligand (EL) dataset, we compared to BigMHC-EL and NetMHCpan 4.1. Finally, during stability training, similarity to the NetMHCstabpan [11] results was assessed using cross-validation.

The BA task results can be seen in Fig 1B showing comparable performance to MHCFlurry, a well validated model for BA prediction. This demonstrates the first stage of training for the ImmugenX pMHC module accurately represents the binding affinity of unseen pMHC complexes. This severs as a basis for transfer learning to additional tasks in the next training steps.

The EL task performance similarly showed comparable performance between the tested models and ImmugenX. MHCFlurry was not assessed on this test dataset due to some overlap between their EL training set and this test set, which was also noted by the BigMHC authors. Overall performance and a breakdown of performance by HLA allele are shown in Fig 1C.

Fig 1D shows a comparison of the performance of ImmugenX on the stability task, both training from scratch and with the EL-trained model as the starting model. Due to the shortage of stability data available, a holdout test set was not practical for this task. Similarly, a direct comparison with the NetMHCstabpan model was not possible since both their model and ImmugenX use all of the available data in training and netMHCstabpan is not currently available in a re-trainable format. We see a similarly high Pearson's correlation coefficient to the netMHCstabpan paper, with a best PCC of 0.88 across all available alleles. The model performed best when pretrained with the BA and EL tasks compared to being trained only on the

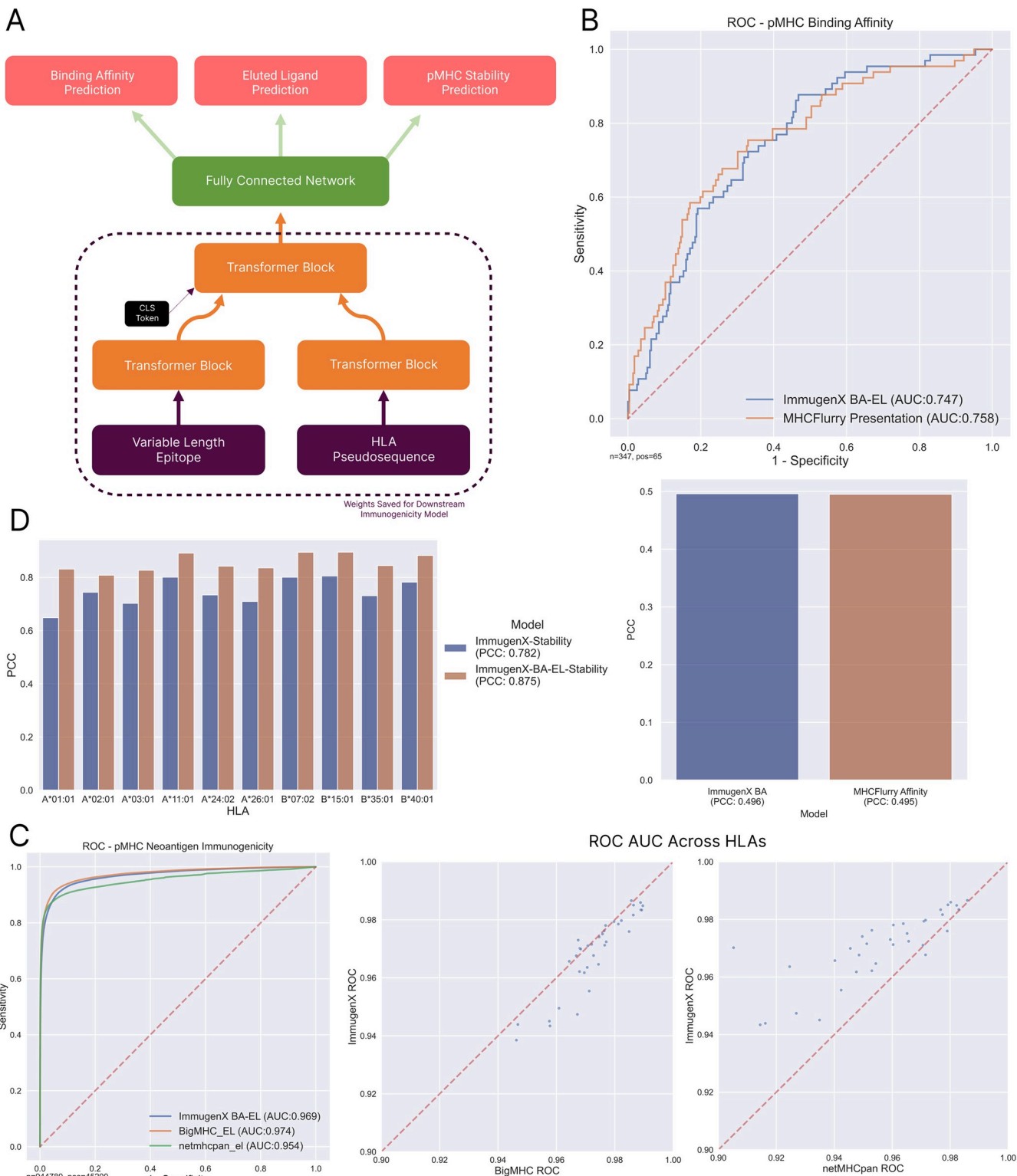

**Fig 1. ImmugenX pMHC pre-trained tasks module.** A: The model architecture of the pMHC module of ImmugenX, trained to perform 3 tasks in series. Weights of the transformer blocks are saved for use in all downstream models for immunogenicity and TCR specificity. B: Performance of pMHC module on the binding affinity task predicting unseen pMHC affinities against MHCFlurry 2.0. Left: ROC curve for predicting strong affinities ($\leq 500nM$), Right: Pearson's correlation coefficient between predictions and measured binding affinities. C: Performance on the eluted ligand holdout set compared against BigMHC and netMHCpan 4.1. Left: ROC curve, Right: Scatter plots for ImmugenX against BigMHC-EL and netMHCpan 4.1 per HLA in the test set. D:

Pearson's correlation coefficient for HLAs in the stability dataset during cross-validation, comparing both the ImmugenX pMHC module trained from scratch and fine-tuned on the BA-EL tasks.

stability data. An improvement in runtime is achieved in ImmugenX compared to NetMHC-stabpan. Benchmark pools of 50,000 pMHC pairs run through ImmugenX's stability module in a uniform 52s on a laptop with an M1 chip and 16GB of RAM, whereas the same pools took between 2–20 minutes with NetMHCstabpan depending on peptide length and HLA diversity.

This iterative training method produced a high performance pMHC module for use in downstream immunogenicity task training.

## 2.2 ImmugenX performs pMHC immunogenicity prediction at state-of-the-art levels on cancer holdout data

The full ImmugenX model for pMHC prediction, shown in Fig 2A, was further trained on CD8+ assay data to predict immunogenicity of a given pMHC pair. Fig 2B and 2C show the receiver-operator characteristic (ROC) and precision-recall curves for ImmugenX compared to its sub-module iterations and other published prediction models on a holdout cancer pMHC immunogenicity dataset. The unseen test dataset composition is shown in 2D. Two versions of ImmugenX were compared, with versions of the pMHC sub-module trained up until the EL task and the full model also trained with stability data. The inclusion of stability pre-training in the pMHC module resulted in the best overall performance (ROC AUC = 0.619, AP = 0.514). Both of the underlying pMHC sub-modules were also tested alone. The EL pMHC sub-module alone performed worst of all of our models, with performance similar to MHCFlurry, indicating peptide elution likelihood was not a highly predictive feature on this dataset alone. NetMHCpan-el is commonly used as a tool for selecting which epitopes to test in immunogenicity assays, leading to a high rate of non-immunogenic but high EL-rank epitopes in the dataset. In the CEDAR portion of the dataset, the mean netMHCpanel rank was 2.02. 2 has been used in other studies as a cutoff for epitopes to test in immunogenicity assays [22] and is recommended as the cutoff for weak binders by the authors [5]. This may be a factor in its near chance performance on this dataset. Stability was an overall better individual feature, with both our stability model and netMHCstabpan [11] outperforming all the BA/EL models. BigMHC [8] performed the best of the non-ImmugenX models run on this dataset. Immunogenicity-specific training was found to be important in discriminating immunogenic from non-immunogenic pMHC pairs in this dataset.

## 2.3 TCR sequence inputs can be integrated into ImmugenX and used to perform TCR specificity prediction

Prediction of TCR specificity from triplets with unseen epitopes has been shown to be limited in recent studies and reviews [18, 23, 19], with results showing above chance performance can be explainable by data leakage from training sets or underlying frequency shifts between data sources [15, 16]. TCR information may still be useful in immunogenicity prediction. To demonstrate the utility of ImmugenX as an architecture for immunogenicity prediction utilising the full peptide-MHC-TCR triplet input, we initially compare it to two recent TCR specificity models using data splits provided by the authors to provide a fair comparison. These experiments have the goal of demonstrating ImmugenX as a model design is capable of using TCR information in its predictions. TCR information is integrated into the model by adding TCR sequence data, as encoded by TCR chain-specific encoders trained in a self-supervised

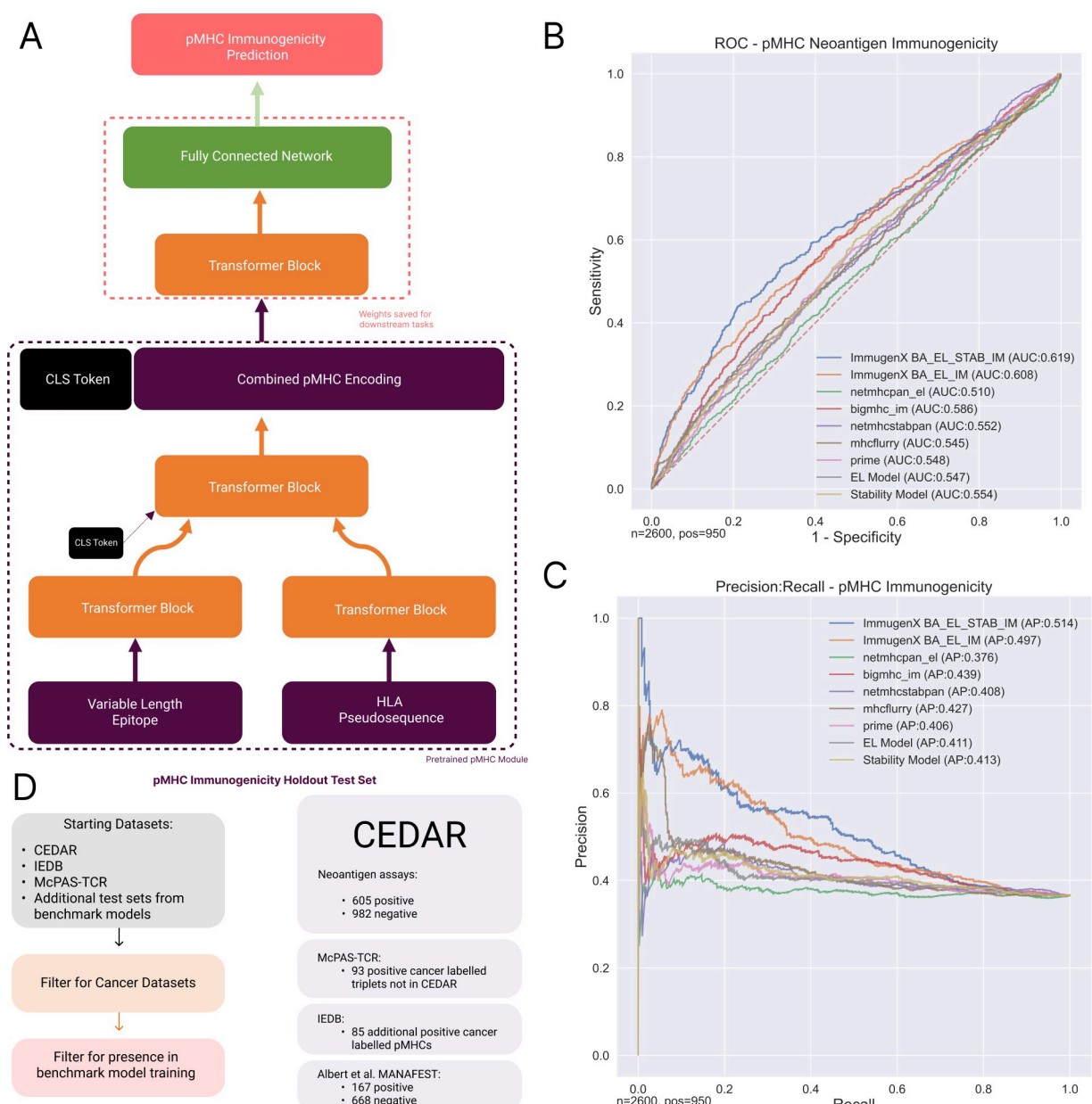

**Fig 2. pMHC immunogenicity prediction with ImmugenX.** A: Architecture of ImmugenX-pMHC. The pretrained pMHC module provides a combined encoding of the pMHC for input to an immunogenicity classification transformer module which is trained on immunogenicity datasets. B: ROC curve on the cancer immunogenicity holdout test set against other modelling approaches. The EL and Stability pMHC modules for ImmugenX are shown also. ImmugenX is shown trained with both of these as a base module. C: Precision recall curves for the holdout test dataset. D: Composition and filtering steps for the holdout cancer test set. CEDAR is taken with supplementary datasets from both non-overlapping data points in other public data sources and test sets from compared models' publications not already in larger public databases.

manner. Fig 3A and 3B show the ImmugenX configuration for full-length paired alpha-beta TCR used in these experiments.

Initially, ImmugenX was compared to NetTCR 2.1 [17], a 1D convolutional neural network-based approach to TCR specificity. Fig 3C and 3D show the performance compared to ImmugenX when trained on the same datasets on the holdout 6th fold as described in the

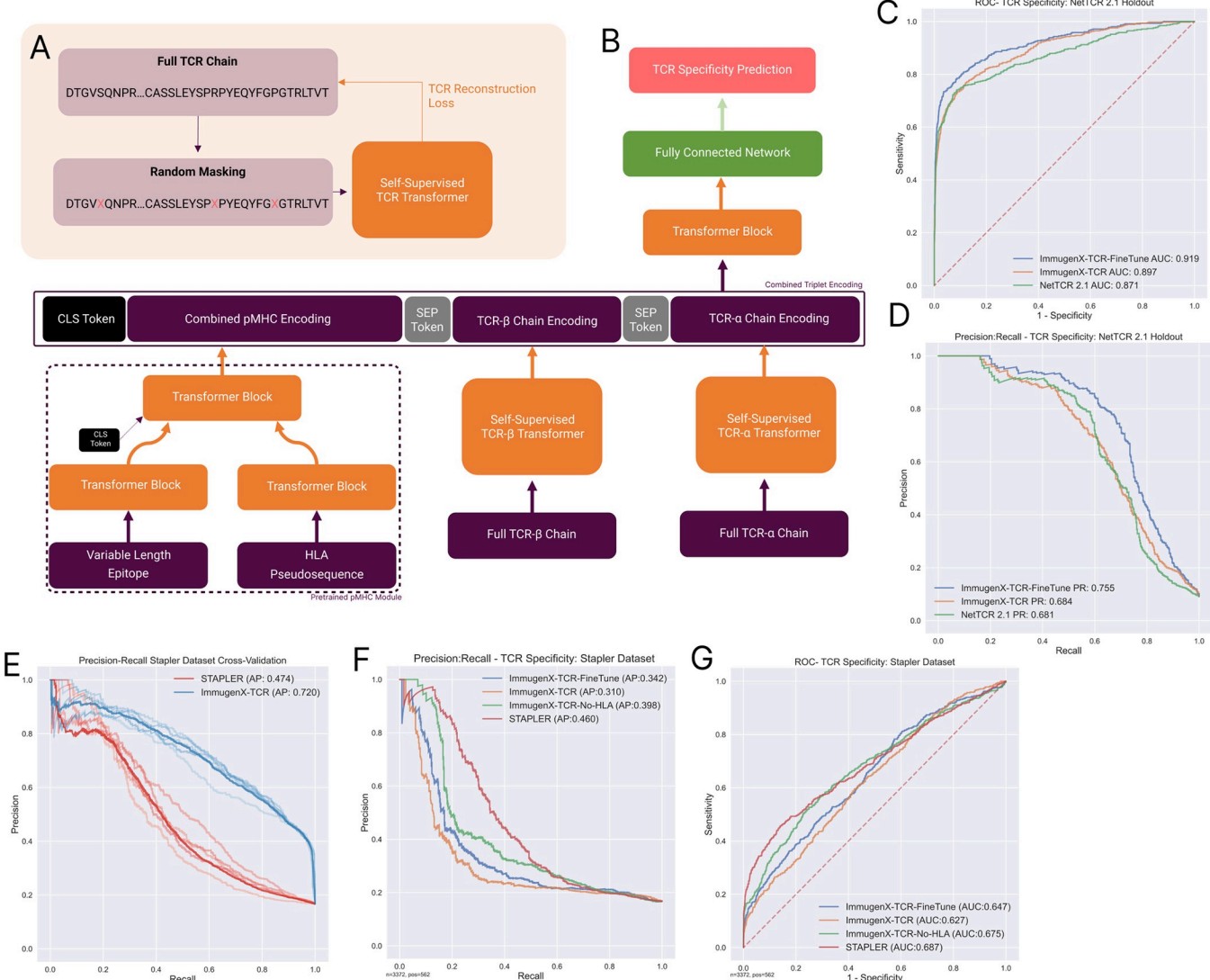

**Fig 3. The method for integrating TCRs into ImmugenX predictions evaluated by performance on TCR specificity tasks.** A: Masking method to train a TCR chain encoder, performed iteratively first for beta chains and subsequently fine-tuned to process alpha chains. B: Architecture for integrating TCRs into ImmugenX predictions by concatenating module outputs into a single sentence to be processed by the prediction head. C and D: Performance of ImmugenX on the NetTCR 2.1 dataset task. Both models are trained on a dataset with 5-fold cross-validation using folds provided by the authors and tested on the 6th fold (results shown). E: Cross-validation performance of both STAPLER and ImmugenX on the training dataset provided by the authors. F and G: Holdout test performance of ImmugenX on the STAPLER holdout set after training using their dataset. An additional version of ImmugenX without HLA inputs is shown to demonstrate the negative impact of HLA inputs in this data split.

paper. NetTCR was retrained using the codebase provided by the authors. Two Versions of ImmugenX were compared, both with frozen encoder weights and with fine-tuning allowed on the whole model. Both versions of ImmugenX demonstrated good performance at this TCR specificity task, with some performance improvement over NetTCR 2.1. The fine-tuned version of the model performed best, indicating increased fine-tuning of the encoder sections is required for TCR specificity tasks.

Fig 3E–3G shows the performance compared to the STAPLER [16] model using their train-test split. Their"VDJdb+ with external negatives" was used as the test set. ImmugenX performed better at the cross-validation task with an improved average precision but had reduced

performance on the test set compared to STAPLER. The main difference between these two models is the inclusion of HLA inputs in ImmugenX, since STAPLER is based on the BERT [24] design which is overall similar to ImmugenX. A masking experiment was conducted, removing the HLA inputs of ImmugenX and retraining it to identify the influence of this difference on performance. This produced a reduced cross-validation average precision of 0.46, very similar to the result in the STAPLER paper of 0.47. On the test set ImmugenX had a lower performance compared to STAPLER showing a reduced overall average precision. Again, there was an improvement in performance with fine-tuning allowed across the network. Here the HLA masking experiment showed the difference in performance was greatly reduced by removing the HLA element of ImmugenX and retraining. HLA bias is a known issue in TCR specificity datasets [19], with 98% of the STAPLER test set being composed of three HLA alleles and 63% A02:01. This means the HLA residues, which the ImmugenX design is designed to use, explain little variance in this dataset. While the training set is similarly biased (59% A02:01) there is an improvement in cross-validation precision when the HLA pseudosequence residues are available to the model.

These experiments demonstrate ImmugenX is able to use TCR information to perform prediction tasks at a comparable level to alternative designs seen in state-of-the-art models.

## 2.4 TCR inputs can improve the performance of ImmugenX' immunogenicity prediction

The previous experiments demonstrated ImmugenX as an architecture for TCR specificity tasks, showing its ability to use information from TCR-based inputs. An additional experiment was conducted to investigate whether having patient TCR information would assist in predicting the immunogenicity of a pMHC complex. A reduced holdout test set from CEDAR was used to test the model where paired CDR3-beta chain information was available as input for ImmugenX. Negatives were constructed using pMHCs marked as negative on TCR assays in CEDAR paired with TCRs from the positive set (Fig 4A–4D). Fig 4E and 4F show an improved performance was achieved by including the TCR information in the model input compared to the pMHC-only model.

While there is limited data available in the test set, these results indicate that improved neoantigen ranking could be achieved when a TCR context is available as model input.

## 2.5 ImmugenX model performance is informed by underlying learned features and position-aware residue importance

pMHC encoding features were analysed for their relation to underlying physiochemical features, such as those that could be used in an immunogenicity model with engineered features. Fig 5 shows a T-distributed stochastic neighbour embedding (t-SNE) of the encoded output of the CLS token and across the peptide tokens as output by the pMHC encoding module of ImmugenX. An alternative PCA based version is shown in S5 Fig. These are shown both for distributions of physicochemical properties of the peptide and input HLA alleles. The pMHC immunogenicity test set was used for this analysis. A support vector regression was performed on this dataset to demonstrate the extent to which these features could be recovered from the encoding at this stage of the network. The GRAVY index score, a measure of the overall hydrophobicity, for the input peptide is well represented in both the CLS token and peptide embedding. The instability index [25] a more complex calculated metric of the peptide was not well represented in the embedding. The secondary structure fractions of helix, turn and sheet were all represented well in the protein encoding seen by the immunogenicity prediction heads. The input HLA allele is visually identifiable in the t-SNE plots for both the CLS token and

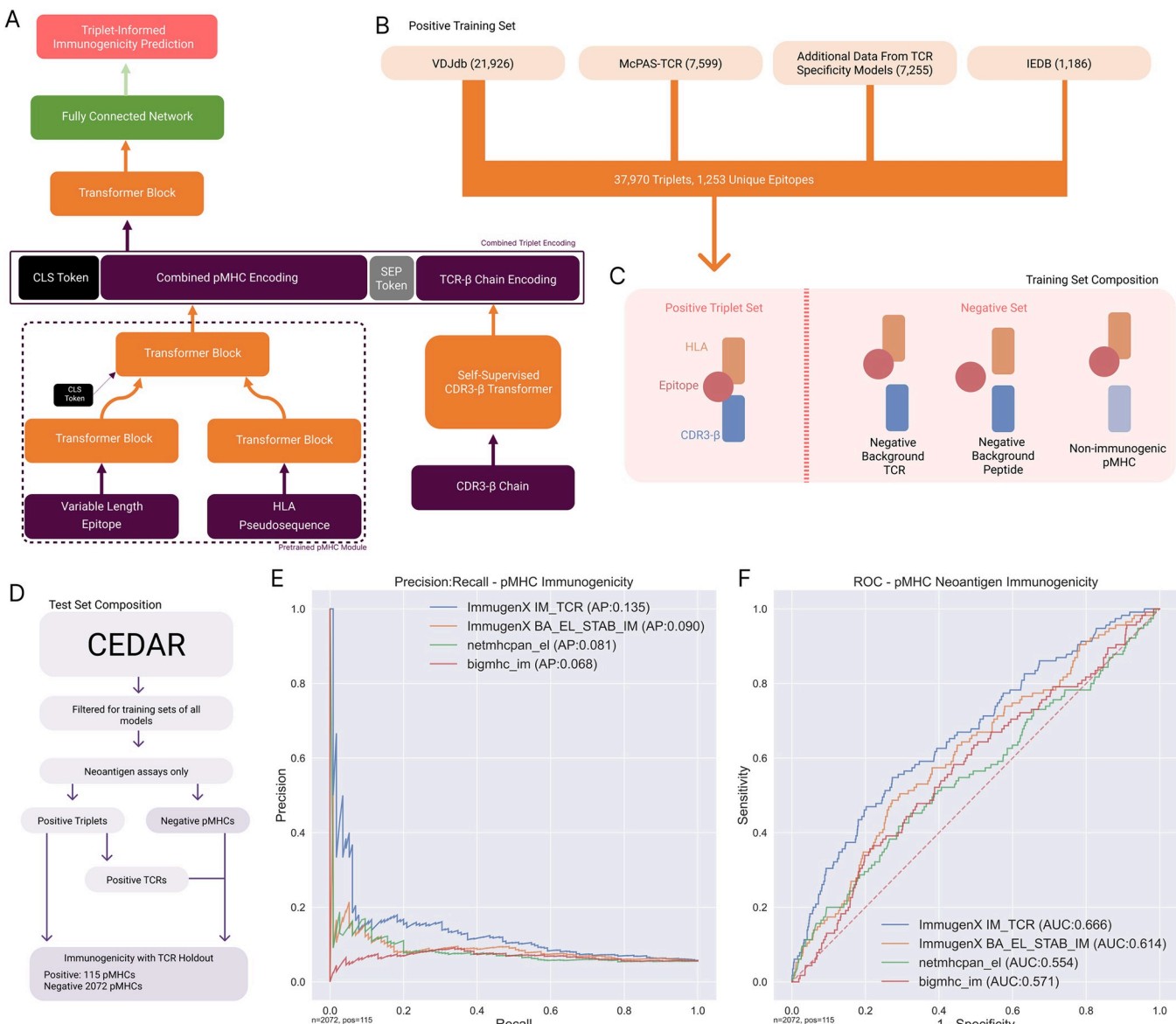

**Fig 4. pMHC immunogenicity prediction with additional TCR information.** ImmugenX makes pMHC immunogenicity predictions in a given TCR context. A: Architecture for integrating CDR3 sequences into ImmugenX predictions, utilising a pretrained self-supervised CDR3 encoder. B: Training dataset sources for fine-tuning the immunogenicity prediction head of ImmugenX. C: Composition of the fine-tuning dataset, utilising 3 types of negative data: positive pMHCs paired with negative background TCRs; Correctly matched MHC-TCR pairs with negative background wild-type peptides from the human proteome; Non-immunogenic pMHCs from TCR assays paired with TCRs from the positive training set. D: Composition of the neoantigen holdout test set from CEDAR, restricted to positives with known TCR triplets. Negative data points were created using negative pMHCs paired with TCRs from the positive set. E and F: Performance curves for ImmugenX with and without the TCR information input, along with netMHCpan 4.1 and BigMHC.

peptide tokens, indicating there is a good integration of the HLA into the classification input as well as via cross-attention applied to the peptide encodings.

Importance analysis was performed using the SHAP python package [26] using a permutation approach with the training set as the background distribution to sample from. The results are shown in Fig 6. The stability-trained pMHC module showed high SHAP values in both the common anchor residue locations and a C-terminal usage separated per n-mer length. In Fig 6H the HLA-A*02:01 9mer samples showed clear impacts from changes to the anchor

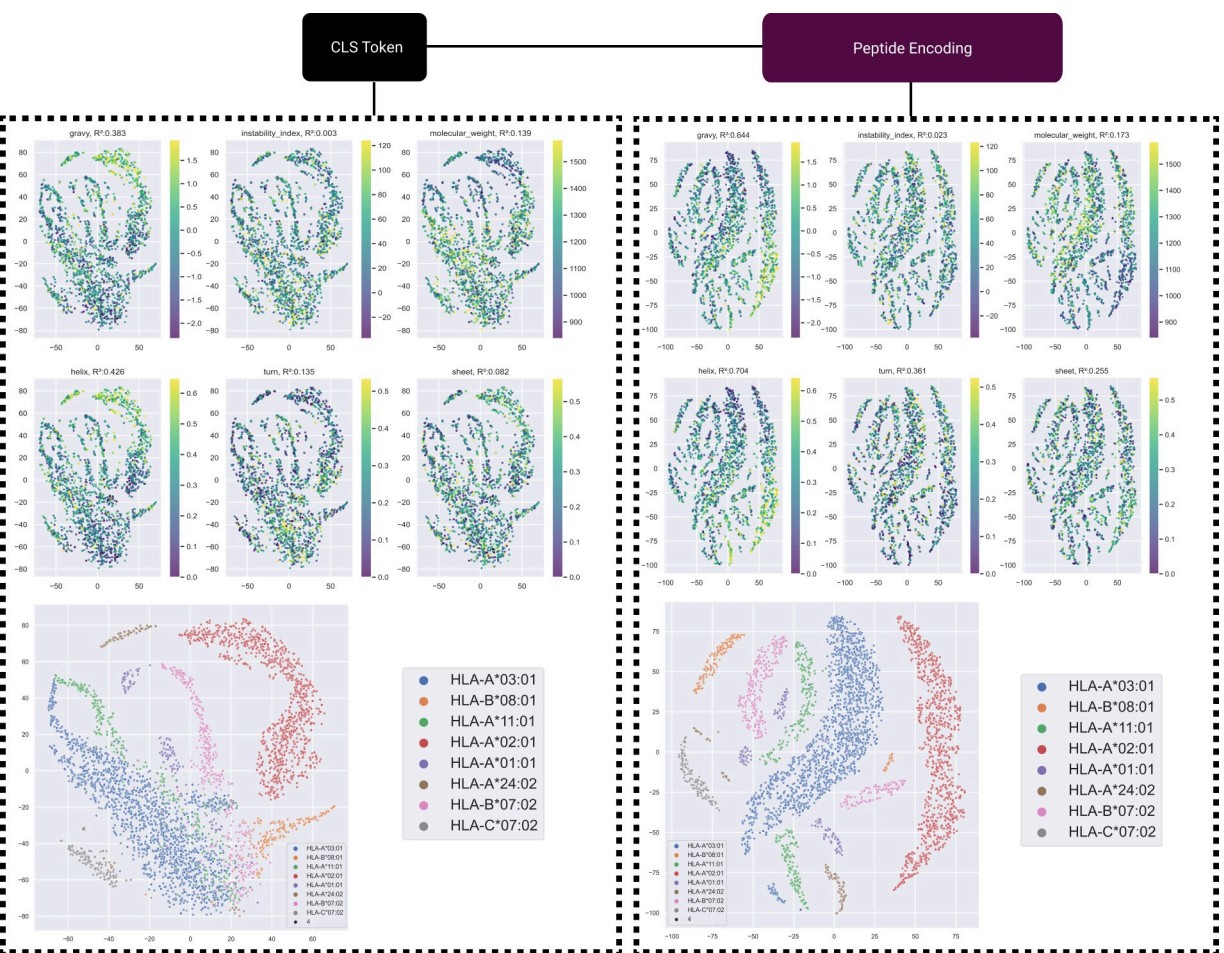

**Fig 5. pMHC encoding representations of physiochemical properties.** t-SNE visualisations of both the CLS token and all peptide tokens coloured for the physiochemical properties of the input peptide. $R^2$ values are shown for predicting the true value of each metric with a support vector machine model using the encodings as feature inputs. The bottom plots show the respective t-SNE plots coloured for the 8 most frequent HLA alleles, demonstrating separation between encodings for some HLAs both in the CLS token and peptide token encodings.

positions for that allele at positions 2 and 9. This pattern was replicated in the full ImmugenX immunogenicity model but with lower differential SHAP values between residues, indicating a greater relative usage of non-anchor location residues for determining immunogenicity. Individual examples of importance on known structures is shown in the S1 Text. The Immu-genX-TCR model was also analysed to determine which aspects of the TCR input were used and how much it factored into predictions. Fig 6 shows a similar distribution of residue usage to those previously reported by Dens et al. for the CDR3-$\beta$ [27], with a core region from residue 4–15 accounting for most of the residue importance in our analysis. We found that while the TCR currently does impact the immunogenicity prediction, it is the least used of the three input sequences when performing the pMHC-TCR immunogenicity task.

## 2.6 Variances in performance between models shows the characteristics of difficult to predict peptides

Additional analyses were performed to assess sources of difference between ImmugenX and other models on the pMHC immunogenicity task, shown in Fig 7. The ImmugenX stability

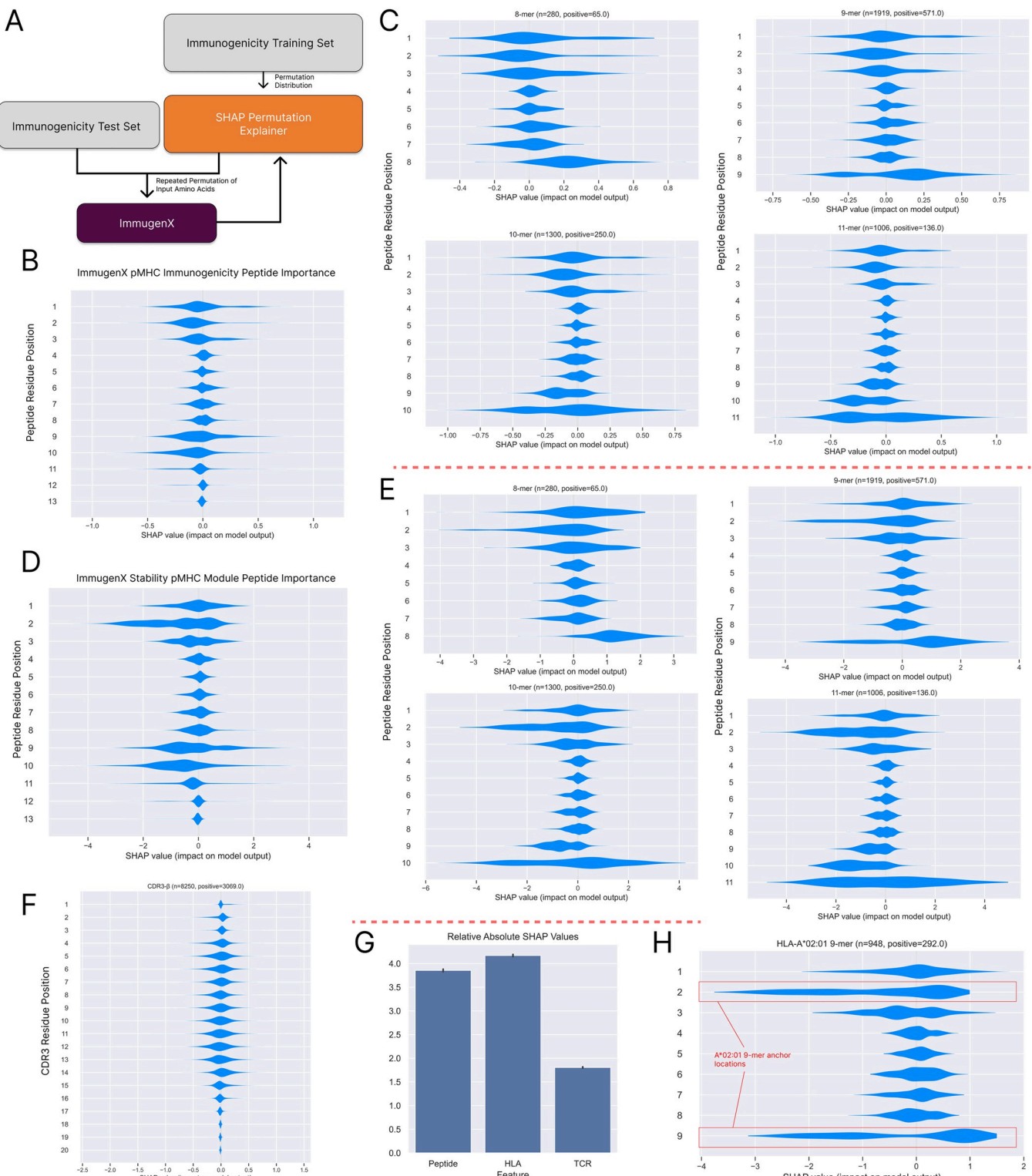

**Fig 6. Input residue usage by ImmugenX from estimated SHAP values.** A: SHAP values by masking the test inputs based on the background distribution of inputs calculated by the training set. Over permutations of masking input values the importance of each input residue to ImmugenX can be estimated. B: Peptide residue importance for the pMHC immunogenicity model across the immunogenicity test set. C: Peptide residue split by peptide length for the 4 most prevalent lengths. D/E: Peptide residue importance for the stability pMHC base module across all data and the most common peptide lengths. F: CDR3 residue usage for the TCR-Immunogenicity prediction model. G: Relative mean importance for all residues in the TCR-Immunogenicity model across peptide, HLA and TCR fractions. H: SHAP values for HLA-A*02:01 9mers for the stability pMHC module, demonstrating specific importance applied to the established anchor residues at positions 2 and 9.

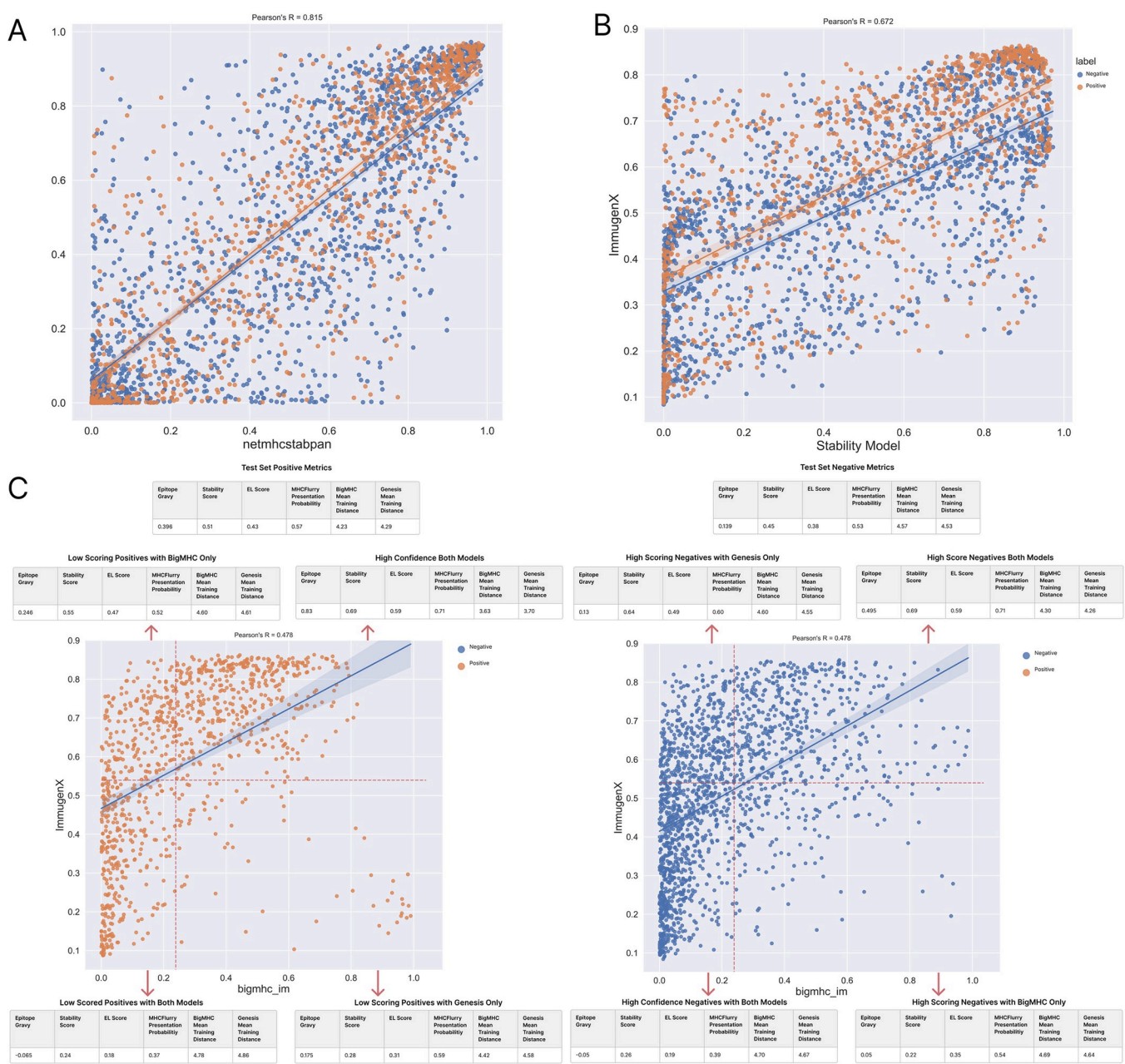

**Fig 7. Model Performance Analysis on pMHC Test Set.** A: Scatter plot of the ImmugenX stability pMHC sub-module and NetMHCstabpan demonstrating high agreement on the test set along with a large population of positive data points with low predicted stability by both models. B: ImmugenX scores plotted against its stability-trained pMHC sub-module. The improved performance was achieved by the rescue of low-stability pMHCs after training on the immunogenicity task. C: ImmugenX and BigMHC scores scatter plot. The Red dashed line shows the mean scores for both models on the whole dataset. Break out tables showing metrics for positive and negative samples separately in each quadrant indicating shifts in mean values for correctly and incorrectly predicted pMHCs, along with differences between the models.

pMHC module produces scores which are highly correlated with netMHCstabpan [11] (Pearson's R = 0.806), a widely used pMHC stability prediction tool. This is expected as they share a training dataset but differ on pre-training steps and architecture. Fig 7A shows a large population of positive pMHC complexes from immunogenicity assays with low predicted stability. The full ImmugenX model trained at the immunogenicity task shows a shift in scores

compared to the stability pMHC module, rescuing some lower predicted stability data points (Fig 2)). ImmugenX and BigMHC immunogenicity scores were analysed to investigate data points on which the models disagreed, as well as the characteristics of false positives and negatives. Scores from our stability and EL-trained pMHC modules, along with MHCFlurry presentation probability were calculated. Distance from the respective model training sets was measured by calculating the minimum Levenshtein distance to any epitope in the training data. The GRAVY score of the peptides was also calculated to determine the mean hydrophobicity of the populations. The dataset was split into quadrants based on the two models' mean scores across the whole dataset. Positive data points with high ImmugenX but below the mean BigMHC scores had a higher than average stability score but lower than average MHCFlurry scores. This indicates some false negatives for a model like BigMHC could be rescued by the inclusion of stability training. There was a reverse of this in the quadrant of higher BigMHC and low ImmugenX positives where lower stability and high MHCFlurry were found. Both models performed poorly on positive data points with low scores from our stability and EL pMHC modules, along with low MHCFlurry predictions, low GRAVY and a high distance to training sets. True positives found by both models had higher than average scores on all the metrics.

Both BigMHC and ImmugenX use immunogenicity-related pMHC tasks in their training. While both models are able to improve on predictions of EL and stability alone, neither model generalises well to the populations of positive immunogenic pMHCs which have relatively low scores across EL and stability.

## 3. Discussion

CD8+ neoepitope prediction plays an important role in the development of personalised cancer therapeutics, potentially allowing for highly targeted and more effective treatments. In this work, we've described ImmugenX, a modular language model which achieves state-of-the-art performance on pMHC immunogenicity prediction and can integrate new inputs such as TCRs when available.

The phased pre-training approach detailed here allows for the integration of multiple pMHC complex-specific tasks in the model weights, providing a useful embedding for downstream pMHC immunogenicity modelling at state-of-the-art performance levels. This was shown with a strict separation of datasets, ensuring all test peptides were unseen by the model in any previous task at every point of training. pMHC complex stability in particular was found to be an important additional embedded feature for improved performance on the test set in this work. This aligns with previous work showing pMHC stability's relationship to CD8 + reactivity prediction [11, 13]. Previous models showed that fine-tuning from an EL model [8] provides a strong basis for the more specific task of immunogenicity prediction. Similarly, the output from an EL and stability models have been used as separate inputs to machine learning methods [13, 14] and neural networks [7] previously. Our protein language modelling approach has the advantage of not just integrating other model outputs, but additional features which are observable in the underlying embeddings before the immunogenicity model head. The phased training approach used in ImmugenX allows for the possible inclusion of TCR data, without being limited by the low volume of paired peptide-TCR data available for training an end-to-end model.

The ImmugenX architecture has the advantage of being extendable, both to additional pre-training-based embeddings through new pMHC tasks and to additional feature inputs such as TCR sequences. Paired pMHC-TCR triplet data is currently severely limited, especially when it comes to epitope and HLA diversity [19]. Due to the limited datasets available, TCR

specificity prediction in entirely novel epitopes has not been shown once dataset biases have been sufficiently accounted for [16, 15, 28, 18]. ImmugenX as an architecture demonstrated strong performance at two TCR specificity tasks from previous publications [17, 16] indicating that it is highly capable of integrating TCR inputs, including up to full chain alpha and beta chains, into predictions. We show that integrating CDR3$\beta$ chains into an immunogenicity prediction task can provide an improvement in performance in the limited set of test neoantigens currently available. We found our setup of different pretraining steps and diverse negative data generation was important in achieving this increase. TCR input was found to have the least impact on the overall prediction score; however, the shift in performance seen here indicates the patient TCR repertoire could be an important input in improving neoepitope ranking. While these results are limited by available data, this work demonstrates the potential gains from integrating TCR information into immunogenicity prediction.

The extendable design of ImmugenX allows for additional features to be integrated by adding to the feature sentence structure as shown with TCRs. Features which have been established to impact immunogenicity, such as expression levels [13] and antigen clonality [29] could similarly be appended to the ImmugenX inputs. Currently, the limiting factor is data availability to support fitting and testing of the model in these alternative configurations. Public datasets rarely include these patient-level features. In our S1 Text we investigate the usage of a large pretrained model, ESM2 [30], to replace the embedding layers of our model. This experiment showed there was no benefit with current datasets from using this embedding compared to our iterative training methods alone. Similarly, structural features may improve the performance in future, especially when considering TCR inputs; however, the low volume of known triplet structures makes this challenging [31, 18]. Ongoing research efforts, such as the MATCHMAKERS grand challenge, aim to improve the availability of structural data. As more data becomes available, additional features can be added to ImmugenX either as additional modules or embedding layer features.

The SHAP analysis of ImmugenX showed a more balanced use of residues in the immunogenicity-trained full model compared to the underlying pMHC sub-module trained on the BA, EL and stability tasks which was highly impacted by the associated HLA binding motifs. This is the expected behaviour as the residues other than the anchor locations are involved in TCR binding. This result indicates ImmugenX is learning positional importance which has been found to be beneficial when explicitly encoded in models like PRIME [7]. The current version of the model places more positional importance on known biologically relevant locations; however, future version of the model will need to be reassessed in this manner to ensure spurious features aren't learned from any new datasets, which may not be apparent from benchmark scores alone.

Much like the models directly compared here and other published models, this study is limited by the availability and composition of immunogenicity data. The proportion of explained variance derivable from either engineered features or learned features from existing training data provides limited overall performance. In particular, there is a disconnect between BA or EL predictions and CD8+ reactivity based on data deposited in public databases such as CEDAR. There exists a potential bias in reactivity datasets due to the use of BA/EL models to determine test candidates from called mutations [13, 14, 22], since these tools have biases associated with their training data acquisition methods [32, 33]. There is also a significant issue in the field with possible false negatives in datasets, with highly differential reactivity between patients [34]. We found the recall of immunogenic epitopes with low average scores on associated features such as EL rank and stability was poor in both ImmugenX and BigMHC, the next best model tested. For this population of antigens, performance is unlikely to be greatly improved by better-engineered models alone. Larger improvements require much more

reactivity data, reduced bias in selecting what is tested and additional features beyond the pMHC sequences. Our work indicates paired TCR datasets in particular would be highly valuable for both improving epitope level immunogenicity prediction and providing a patient-specific context to the model.

In our tests EL models perform very poorly on immunogenicity prediction due to the bias in which epitopes are tested being skewed to highly ranked epitopes; however, these tools are still likely useful for minimal epitope selection compared to randomly selected epitopes from a given mutation. Additional datasets to understand the population of low EL rank but positive reactivity epitopes in the datasets would be highly valuable.

ImmugenX provides a strong framework for immunogenicity prediction using a modular protein language model which achieves state-of-the-art level performance at predicting immunogenicity on publicly available CD8+ reactivity data.

## 4. Methods

ImmugenX is designed as a modular protein language model, trained in an iterative manner using different transfer learning tasks. The model is broken up into different reusable modules depending on the end prediction goal and the available inputs. All models were developed in Python 3.8.11 using PyTorch 1.12 [35]. Training was performed on NVIDIA Tesla T4 GPUs with 16GB of VRAM. All models were trained with the Adamax optimiser to minimise binary cross-entropy loss across all classification and regression tasks.

### 4.1 Data-leakage control

ImmugenX is trained in an iterative manner and uses cross-validation in some optimisation and benchmarking experiments at different stages. To ensure a clean data split and no information leakage occurred between tasks, a filtering strategy was applied to all datasets iteratively to remove any data previously seen by the model. Taking the pMHC immunogenicity dataset and working backwards through training steps, data points were removed if there was an 8mer level match with any peptide used in a downstream task. This ensured all test sets were completely unseen by the model at any previous training step, resulting in a small reduction in the training sets for some tasks while preserving the size of the limited test sets of unseen data. For all 5-fold cross validation experiments peptides were assigned to the same folds as any 8mer matched peptides to ensure highly similar peptides were never used in both a training and testing set. S1 Fig provides a diagram of the strategies used here.

### 4.2. Peptide MHC encoding module

The initial pMHC module is trained to perform a binding affinity task, followed by an eluted ligand prediction task and finally fine-tuned on a pMHC stability task. The model architecture and training parameters were selected based on 5-fold crossvalidation performance on the EL task. The design presented here is an encoder-only protein language model with separate transformer [36] branches for both the epitope and HLA pseudosequences, which are then concatenated along with a classification token before another layer of transformer encoders. Each transformer block is made up of 4 encoder layers. Early developmental experiments investigated deeper transformer blocks of 6 and 8 layers, but found performance was maximized with 4 layers. For pMHC training tasks the classification token is fed into a fully connected network for final classification. MHC pseudosequences are based on the MHCFlurry 2.0 alignment provided with their publication [6]. Amino acid sequences are encoded by branch-independent embedding layers with an embedding depth of 72.

For the binding affinity training step of the pMHC module, the binding affinity portion of the MHCFlurry 2.0 training set was used [6]. This initially comprised 219,596 affinity measurements, filtered for only human quantitative measurement data, resulting in a dataset of 99,245 measurements. Binding affinity measurements, in IC50 values, were scaled to between 1–0 using the formula $x = 1-log(aff)/log(50,000)$ [37]. Initial hyperparameter tuning was performed with 5-fold cross-validation. Test performance, to ensure the model performed well for unseen binding affinity prediction, was assessed using an IEDB [21] search for binding affinity data with IC50 quantitative values deposited after January 2021. This ensured a direct comparison to MHCFlurry was possible as this data had no overlap with their training set. This search yielded 427 pMHC complexes, with 105 entries having a binding affinity under 500nM. Performance was compared as both Pearson's correlation coefficient to the measured binding affinity value and performance at classifying presented values, using 500nM as the positive cutoff point.

The EL task was trained using the BigMHC-EL training dataset which was a combination of the MHCFlurry 2.0 [6] and NetMHCpan 4.1 [5] single allelic EL datasets. Model layer depth, batch size and learning rate were optimised using 5-fold cross-validation on the 16,992,037 (240,565 positive) pMHCs. The BigMHC-EL [8] test set was used for performance evaluation at the EL task covering 900,592 negative and 45,400 positives, also taken from NetMHCpan 4.1 but with duplicates removed and filtered by our filtering strategy (S1 Fig). All this data was originally acquired through immunopeptidomics studies with binary labels for whether a given pMHC was eluted.

The BA-EL pMHC model was then trained further to perform a pMHC complex stability task. 28,088 half-life measurements were provided by the authors of the NetMHCstabpan paper [11] and one additional study of yellow fever vaccine epitopes [38]. The raw data is scaled in the same manner as demonstrated in the NetMHCstabpan paper, $S = 2^{-t0/hl}$, where S is the converted value and $t_0$ is the conversion constant. The constant was set to 1, which was determined to be a reasonable pan-allotype value in the original paper and which was confirmed in our early developmental experiments. Negative entries were added from the MHCFlurry 2.0 dataset [6] where the measurement value was over 20,000nM. 1000 negative samples per MHC allele represented in the positive set were sampled. Performance was assessed by Pearson's correlation coefficient to the actual half-life values after 5-fold cross-validation.

### 4.3. Peptide MHC immunogenicity prediction

For pMHC immunogenicity prediction the final transformer block outputs from the pMHC module are provided as the input. The weights of the pMHC module are frozen during training of the immunogenicity head layers. An additional transformer and classification head are trained on immunogenicity-specific data. Training data was compiled from publicly available data sources including both positive and negative epitopes: IEDB [21], VDJdb [39], TESLA [2], McPAS-TCR [40] and the PRIME model training sets [7]. Additional negatives were sampled from the human proteome using the consensus coding sequence project (CCDS) [41].

For benchmarking ImmugenX for pMHC immunogenicity prediction two variants were compared. One with the pMHC module trained up until the EL task and another with the full training including the stability prediction task to assess the utility of this additional training task to immunogenicity. ImmugenX was compared to pMHC binding/elution models netMHCpan 4.1 [5] and MHCFlurry 2.0 [6], the pMHC stability model netMHCstabpan [11], along with the pMHC immunogenicity models PRIME 2.0 [7] and BigMHC [8]. The EL and EL-Stability pMHC sub-modules of ImmugenX were also compared.

The holdout test set was compiled by combining the cancer-specific dataset CEDAR [42], unseen cancer-specific pMHCs from IEDB and McPAS-TCR and the holdout MANAFEST assay dataset of 16 cancer patients from BigMHC [8]. CEDAR was searched for t-cell assays in humans only. The full set was filtered to remove peptides contained in the training set for the immunogenicity comparison models BigMHC and PRIME.

This resulted in a ImmugenX training set of 9044 pMHCs (2588 positives) and a test set of 2600 pMHCs (950 positives). This set was the start point for our data filtering policy for all datasets, shown in S1 Fig.

## 4.4 TCR specificity prediction

ImmugenX is designed as a modular language modelling approach for immunogenicity prediction, with the ability to add additional features to the input of the classification head. TCR specificity as a task is performed using peptide:HLA:TCR triplets as input. The pMHC encoding module is unchanged to process the peptide:HLA paired input, with the immunogenicity classification head fine-tuned with the addition of TCR-based inputs. For these benchmarking experiments full-length TCRs with both alpha and beta chains were used to compare to existing state-of-the-art models; however, other configurations such as CDR3-beta only input are also possible with ImmugenX.

A set of TCR encoding modules consisting of a 2-layer transformer encoders were pre-trained in a self-supervised manner using a random masking approach, where at each step 13% of the input amino acids are randomly masked to the model. TCR-beta chains were generated from datasets available from a sequencing dataset of 666 patients available from Adaptive Biotechnologies [43]. Full-length amino acid sequences were reconstructed using the V and J allele annotations, with sequences defined in IMGT/GENE-DB [44]. Entries containing ambiguous residues or nonstandard amino acids were removed. 15,363,111 unique TCR-beta sequences were used to train the base model, with 5% randomly selected to use as a validation set for hyperparameter optimisation. For alpha chain inputs a specific encoder was trained by fine-tuning the beta chain encoder with the alpha chain pre-training set from the STAPLER model [16], resulting in 46,207 unique alpha chains after processing. Both models were trained using negative log-likelihood loss of reconstructing the original sequence from the masked input using a projection back to amino acid space using a fully connected layer, as shown in Fig 3A.

The final combined model is constructed by combining the outputs of the pMHC and TCR modules, separated by special separation tokens. As with the pMHC immunogenicity model, a classification head is provided with the combined outputs of the preceding encoders. Fig 3B shows this configuration of ImmugenX. This is fine-tuned using the pMHC immunogenicity prediction head. For the TCR specificity task, versions of ImmugenX were trained with either encoder weights frozen as with the pMHC immunogenicity task, or the entire model was fine-tuned including the encoders.

NetTCR 2.1 [17] is a 1D CNN-based model which processes the TCR and peptide separately before concatenating the outputs for input to a set of fully connected layers. To include measured negatives in the datasets rather than only using mixed negatives, they focused on 6 peptides for which there were negative TCRs in the 10x Genomics Single Cell Immune Profiling study. Additionally, they limited the search to only samples where all CDR regions were available for both alpha and beta chains, which has been found previously to improve TCR specificity prediction. This produced a dataset of 2,541 positives along with 12,848 true negatives from 10x and 12,705 randomly swapped negatives. The cross-validation and test splits were used as provided by the authors. For ImmugenX input, the HLAs were acquired from the

original data sources. These were randomly sampled from negative samples, meaning epitopes were always presented with a known positive HLA.

STAPLER [16] is a transformer-based model based on the BERT [24] architecture, reading full-length TCR alpha and beta chains along with the epitope of interest as a sequence of tokens. Their model is trained with a mixture of masked language modelling and fine-tuning tasks for epitope-TCR pairs. The comparison task with ImmugenX is performed using the data provided from their GitHub repository consisting of a fine-tuning training set and a hold-out test set consisting of positive triplets from VDJdb [39], along with a subset from their other sources and externally sampled negative TCRs. Positives were a mixture of seen and unseen epitopes. They identified internal shuffling of the VDJdb TCRs to be a source of data-leakage, advising the use of the external TCRs. This provided a fine-tuning training set of 23,410 triplets and a test set of 3,372 triplets (562 positive).

## 4.5. TCR Assisted immunogenicity prediction

While TCR specificity on unseen episodes is currently not possible with existing datasets and architectures, TCR data when available could be used in ranking pMHC complexes for immunogenicity using a compatible framework. The TCR version of ImmugenX was compared to the pMHC-only version to assess if TCR data could be useful in improving immunogenicity prediction when available. For this experiment, only the CDR3-beta chains were used due shortage of datasets with full paired chain information, particularly for holdout pMHCs unseen by other models. This configuration of ImmugenX is shown in Fig 4A.

Training data was composed of positive triplets from the same sources as the pMHC-only experiments, filtered for datasets where there are known reactive CDR3beta sequences. This resulted in a training set of 37,970 positive triplets covering 1,253 unique epitopes. 3 distinct types of negative data were generated as shown in Fig 4C. First, negative wild-type peptides were sampled from the CCDS [41] and paired with TCRs taken from the positive set. Second, positive pMHCs were paired with negative TCRs from the background distribution used in the TCR encoder pre-training. Finally, negative immunogenic pMHCs from the PRIME and TESLA datasets were paired with TCRs from the positive fraction to create likely presented pMHCs but non-immunogenic triplets. These negative types were produced at equal proportions, with a total negative to positive ratio of 100:1. 5-fold cross-validation was used to fit training parameters batch size and learning rate using a grid search approach.

The holdout test set was comprised of positives taken from CEDAR [42], filtered for t-cell assays of neoepitopes in humans and with beta chain CDR3 sequences available and negatives from the same search without the CDR3 requirement. This was a subset of our pMHC immunogenicity test set. Negative samples were paired with the positive TCR set to ensure differences between the positive and negative datasets could not be detected by a distribution shift in the TCR repertoire alone. Any epitopes present in both the training and test sets were removed from the training set for either ImmugenX or the comparison models. To maximise the possible test data, only BigMHC was compared as the best-performing other model from the pMHC-only immunogenicity prediction. NetMHCpan-el was also included to compare against a presentation-only model. This produced a reduced test set of 115 positive pMHCs with at least one known reactive TCR and 1957 negative pMHCs paired with the positive fraction's TCRs. ImmugenX-TCR scores were aggregated per pMHC by taking the maximum predicted score.

## 4.6 Interpretability

Analysis representation of physiochemical property representation in the model embeddings was performed by visualisation using T-distributed stochastic neighbour embeddings (t-SNE)

and training an estimator model to recover these properties from the embeddings. The immunogenicity test set was used for this analysis by passing each data point through the pMHC module and analysing the output of the embedding by the model for use by the immunogenicity head models. 2D t-SNE embeddings were generated using the scikit-learn python package (version 1.0.2) [45]. Physiochemical properties tested were the GRAVY score, instability index [25], molecular weight and secondary structure fractions. These were all computed using Bio-Python (version 1.78) on each input peptide. Support vector regression models were trained to predict the actual values using the embeddings as inputs to determine if these features were represented by the model for immunogenicity prediction input. SVR models were trained using scikit-learn.

Input importance analysis was performed using the SHAP python package version 0.40.0 [26], using their permutation explainer method. Here the training immunogenicity dataset was used as the background distribution for permuting amino acid values. This estimator calculates importance values by computing the change in model output when input values are replaced by those from the background distribution across a large number of iterations. The immunogenicity dataset was again used as the dataset for this analysis. Values were calculated for the stability pMHC module, ImmugenX and ImmugenX-TCR models. The stability and ImmugenX models were investigated for their usage of peptide residues. The TCR-inclusive model was analysed similarly, but also for CDR3-$\beta$ residue usage. Total mean usage across peptide, HLA and TCR inputs was also calculated to show the impact of each module on model predictions. Individual cases for known positive interacting pMHC:TCR pairs are shown in S1 Text.

## Supporting information

**S1 Fig.** Filtering strategies for ensuring no data leakage between training steps, including for the sub-module tasks (A) and the cross-validation strategy ensuring all epitopes with matching 8mer substrings appear in the same fold. All data is filtered based on matches of 8 consecutive amino acids in the peptide.
(TIFF)

**S2 Fig. Performance comparison between main model and ESM based model on the pMHC immunogenicity holdout dataset**
(TIFF)

**S3 Fig. SHAP values for the known interaction between the MART-1 9mer AAGIGILTV presented by HLA-A\*02:01 interacting with the engineered DMF5 TCR.** Structure from Hellman et al. 2019 [4]. High SHAP values shown on the known HLA-A2 anchor locations at residue 2 and the C-terminal. Residues 1, 3 and 5–7 have well established preferred and deleterious amino acids, resulting in high absolute SHAP values. Residues at 3 and 5–7 are in position for importance with TCR binding. CDR3-beta residue importance is focused on regions at the binding interface, with low values at the tails.
(TIFF)

**S4 Fig. SHAP values for the known interaction between the MAGE-A3 epitope EVDPIGHLY presented by HLA-A\*01:01 and a known reactive TCR, from Raman et al.** 2016 [5]. HLA-A1 has a very narrow binding motif, especially at positions 2 and the C-terminal. LY is the most preferred peptide suffix for binding. Core residues on both the peptide and CDR3-beta chain likely to be interacting based on structural distance have above background SHAP values.
(TIFF)

**S5 Fig. Alternative version of Fig 5 with PCA based 2D projection of encodings.**
(TIFF)

**S1 Text. Supplementary Methods.**
(DOCX)

## Author Contributions

**Conceptualization:** Hugh O'Brien, Max Salm, Laura T. Morton, Maciej Szukszto, Pablo D. Becker, Andrew Craig, Yardena Samuels, Charles Swanton, Marc R. Mansour, Sine Reker Hadrup, Sergio A. Quezada.

**Data curation:** Hugh O'Brien, Morten Nielsen.

**Formal analysis:** Hugh O'Brien, Max Salm, Felix O'Farrell.

**Funding acquisition:** Max Salm, Pablo D. Becker, Yardena Samuels, Charles Swanton, Marc R. Mansour, Sine Reker Hadrup, Sergio A. Quezada.

**Investigation:** Hugh O'Brien, Laura T. Morton, Maciej Szukszto.

**Methodology:** Hugh O'Brien, Max Salm, Laura T. Morton, Maciej Szukszto, Felix O'Farrell.

**Project administration:** Max Salm, Pablo D. Becker, Charles Swanton, Marc R. Mansour, Sine Reker Hadrup, Sergio A. Quezada.

**Resources:** Pablo D. Becker, Morten Nielsen, Yardena Samuels, Charles Swanton, Sine Reker Hadrup, Sergio A. Quezada.

**Software:** Hugh O'Brien, Felix O'Farrell, Morten Nielsen.

**Supervision:** Max Salm, Pablo D. Becker, Andrew Craig, Yardena Samuels, Charles Swanton, Marc R. Mansour, Sine Reker Hadrup, Sergio A. Quezada.

**Validation:** Hugh O'Brien, Laura T. Morton, Felix O'Farrell, Charlotte Boulton, Laurence King, Supreet Kaur Bola.

**Visualization:** Hugh O'Brien, Charlotte Boulton.

**Writing – original draft:** Hugh O'Brien, Max Salm, Laura T. Morton, Maciej Szukszto, Felix O'Farrell.

**Writing – review & editing:** Hugh O'Brien, Max Salm, Laura T. Morton, Maciej Szukszto, Felix O'Farrell, Charlotte Boulton, Laurence King, Supreet Kaur Bola, Pablo D. Becker, Andrew Craig, Morten Nielsen, Yardena Samuels, Marc R. Mansour, Sine Reker Hadrup, Sergio A. Quezada.

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
