## [Decision Letter · Decision Letter 0]

29 Jul 2024

Dear Prof Quezada,

Thank you very much for submitting your manuscript "Genesis: A modular protein language modelling approach to immunogenicity prediction" for consideration at PLOS Computational Biology. As with all papers reviewed by the journal, your manuscript was reviewed by members of the editorial board and by several independent reviewers. The reviewers appreciated the attention to an important topic. Based on the reviews, we are likely to accept this manuscript for publication, providing that you modify the manuscript according to the review recommendations.

Sincerely,

Yang Lu, Ph.D.

Academic Editor

PLOS Computational Biology

Amber Smith

Section Editor

PLOS Computational Biology

Reviewer's Responses to Questions

**Comments to the Authors:**

Reviewer #1: Summary

Genesis is a modular protein language model designed to enhance immunogenicity prediction for CD8+ reactive epitopes. It comprises a peptide-MHC (pMHC) encoding module trained on three key tasks—binding affinity, eluted ligand prediction, and stability—alongside optional T-cell receptor (TCR) encoding modules and context-specific immunogenicity prediction heads. The model's interpretability analysis identifies weaknesses and biases in existing datasets, offering a better understanding of immunogenicity features.

Minor Comments

1. For all figures, please increase the font size of x and y axis labels. They are currently too small and difficult to read.

2. What masking rate was used during the pretraining phase?

3. For Figure 5 feature maps, consider including a PCA-based visualization as a supplementary figure, which may provide easier interpretation. In Figure 6 Panel G, the x-axis label is missing for half of the plot.

4. Regarding data leakage control, why did the authors choose to use 8-mer matching? Is there a specific rationale for this choice? Typically, we see controls based on 3, 6, or 9 (3-based) mers. Additionally, have authors tried identity clustering for example?

5. A more comprehensive analysis of position-specific SHAP values would be beneficial. How do these masking changes relate to known TCR contacts or mutations/variants across alleles? Instead of masking, could direct mutation effects be examined?

Suggestions

Note: Suggestions are not formal comments. Authors are not required to address items in the suggestions section, which do not affect the reviewer's decision.

1. Have you considered using a pretrained model such as ESM2 as your backbone? This approach might yield better performance or faster convergence given its inherent knowledge.

2. It is increasingly common to see performance improvements by augmenting protein language models with some degree of structural information, especially for antigens, given the importance of structural complementarity.

3. How would an end-to-end trained model that jointly learns pMHC and TCR features compare to the current approach of separate training?

Reviewer #2: Comment: In 4.2 Peptide MHC Encoding Module, there are 4 encoder layers in the encoder block. Generally protein LMs like ESM [1], ProtTrans [2] and ProteinLM [3] have >= 12 transformer layers to ensure the model can capture enough information.

Increasing the number of layers and hidden size of your model can help with the performance even more.

Question: I wonder have you performed any ablation study on the optimal number of layers in encoder? As well as using general purpose protein language model as the protein sequence encoder, then use the extracted embedding to do the immunogenicity prediction?

[1] Rives, Alexander, et al. Biological structure and function emerge from scaling unsupervised learning to 250 million protein sequences.

[2] Elnaggar, Ahmed, et al. ProtTrans: Toward Understanding the Language of Life Through Self-Supervised Learning.

[3] Xiao, Yijia, et al, Modeling Protein Using Large-scale Pretrain Language Model.

Reviewer #3: The review is uploaded as an attachment.

**Have the authors made all data and (if applicable) computational code underlying the findings in their manuscript fully available?**

Reviewer #1: Yes

Reviewer #2: Yes

Reviewer #3: Yes

PLOS authors have the option to publish the peer review history of their article (what does this mean?). If published, this will include your full peer review and any attached files.

Reviewer #1: No

Reviewer #2: No

Reviewer #3: **Yes: **Ruidong Wu

Figure Files:

Data Requirements:

Reproducibility:

References:

---

## [Decision Letter · Decision Letter 1]

24 Sep 2024

Dear Prof Quezada,

We are pleased to inform you that your manuscript 'A modular protein language modelling approach to immunogenicity prediction' has been provisionally accepted for publication in PLOS Computational Biology.

Best regards,

Yang Lu, Ph.D.

Academic Editor

PLOS Computational Biology

Amber Smith

Section Editor

PLOS Computational Biology

Reviewer's Responses to Questions

**Comments to the Authors:**

Reviewer #1: Thanks for the authors' reply and all revisions. The authors' revisions have fully addressed my comments.

Reviewer #3: Thanks for the response, my questions are well addressed.

**Have the authors made all data and (if applicable) computational code underlying the findings in their manuscript fully available?**

Reviewer #1: Yes

Reviewer #3: None

PLOS authors have the option to publish the peer review history of their article (what does this mean?). If published, this will include your full peer review and any attached files.

Reviewer #1: No

Reviewer #3: **Yes: **Ruidong Wu

---

## [Editor Report · Acceptance letter]

4 Nov 2024

PCOMPBIOL-D-24-00999R1 

A modular protein language modelling approach to immunogenicity prediction

Dear Dr Quezada,

I am pleased to inform you that your manuscript has been formally accepted for publication in PLOS Computational Biology. Your manuscript is now with our production department and you will be notified of the publication date in due course.

With kind regards,

Anita Estes
